# Identification of the Intestinal Microbes Associated with Locomotion

**DOI:** 10.3390/ijms241411392

**Published:** 2023-07-13

**Authors:** Ji-Seon Ahn, Yu-Jin Choi, Han-Byeol Kim, Hea-Jong Chung, Seong-Tshool Hong

**Affiliations:** 1Gwangju Center, Korea Basic Science Institute, Gwangju 61751, Republic of Korea; ajs0105@kbsi.re.kr (J.-S.A.); cyj4854@kbsi.re.kr (Y.-J.C.); gksquf42@kbsi.re.kr (H.-B.K.); 2Department of Biomedical Sciences and Institute for Medical Science, Chonbuk National University Medical School, Jeonju 54907, Republic of Korea

**Keywords:** locomotion, the gut microbiome, fecal microbiota transplantation, FMT

## Abstract

Given the impact of the gut microbiome on human physiology and aging, it is possible that the gut microbiome may affect locomotion in the same way as the host’s own genes. There is not yet any direct evidence linking the gut microbiome to locomotion, though there are some potential connections, such as regular physical activity and the immune system. In this study, we demonstrate that the gut microbiome can contribute differently to locomotion. We remodeled the original gut microbiome of mice through fecal microbiota transplantation (FMT) using human feces and compared the changes in locomotion of the same mice before and three months after FMT. We found that FMT affected locomotion in three different ways: positive, none (the same), and negative. Analysis of the phylogenesis, α-diversities, and β-diversities of the gut microbiome in the three groups showed that a more diverse group of intestinal microbes was established after FMT in each of the three groups, indicating that the human gut microbiome is more diverse than that of mice. The FMT-remodeled gut microbiome in each group was also different from each other. Fold change and linear correlation analyses identified *Lacrimispora indolis*, *Pseudoflavonifractor phocaeensis*, and *Alistipes senegalensis* in the gut microbiome as positive contributors to locomotion, while *Sphingobacterium cibi*, *Prevotellamassilia timonensis*, *Parasutterella excrementihominis*, *Faecalibaculum rodentium,* and *Muribaculum intestinale* were found to have negative effects. This study not only confirms the presence of gut microbiomes that contribute differently to locomotion, but also explains the mixed results in research on the association between the gut microbiome and locomotion.

## 1. Introduction

A typical human has a complex and huge microbial community consisting of 100 trillion microbes in the intestine, known as the gut microbiome [1]. Recent research more and more evidently shows that the gut microbiome plays a significant role in overall health, including digestion, immunity, and even mental health [2,3,4,5,6]. Because of its critical roles, dysbiosis of the gut microbiome can cause various diseases such as inflammatory bowel disease, gastrointestinal tract malignancies, cancer, cholelithiasis, autism, sarcopenia, cachexia, hepatic encephalopathy, allergies, obesity, diabetes, atherosclerosis, metabolic syndrome, Alzheimer’s disease, Parkinson’s disease, etc. [7,8,9,10,11,12,13,14,15,16]. On the other hand, it has been shown that maintaining a healthy gut microbiome has many health benefits [17,18,19,20,21,22]. Although the current research obviously shows that the gut microbiome affects the digestion, immunity, and mental health of its host [23], the effect of the gut microbiome on physiology has not been thoroughly investigated.

Locomotion refers to the ability of an organism to move or travel from one place to another. It involves the physical movement of the body in order to change position or location. Locomotion is a fundamental characteristic of living organisms, enabling them to explore their environment, search for resources, escape from danger, and interact with other individuals. In biology, locomotion can take various forms depending on the organism’s structure, physiology, and evolutionary adaptations. Different modes of locomotion include walking, running, crawling, swimming, flying, hopping, and slithering [24].

There is not yet any direct evidence linking the gut microbiome to locomotion, but there are some potential connections. First, regular physical activity, which often involves locomotion, has been associated with a more diverse and healthy gut microbiome. Exercise and movement can positively influence the composition and diversity of gut microbiota, promoting a balanced microbial community [25,26,27]. Second, both locomotion and the gut microbiome can have impacts on the immune system. Physical activity has been shown to enhance immune function [28], while the gut microbiome interacts with the immune system, influencing its development and response [4,5]. A healthy gut microbiome can help regulate immune responses, potentially affecting an individual’s ability to engage in locomotion without health complications.

Locomotion is a complex trait and can be influenced by various factors, including genetics, physical and environmental factors, etc. [29,30,31,32]. Therefore, studies on the interaction between locomotion and the gut microbiome must be conducted through excluding other environmental or individual factors. Human studies, where individual genetic backgrounds cannot be controlled, have limitations in accurately evaluating the effect of the gut microbiome. On the other hand, mouse studies, where genetic factors can be controlled, allow for a more accurate evaluation of the effect of the gut microbiome through eliminating the influence of genetic factors [33].

In order to overcome the difficulty of separating the effects of genetic factors from the gut microbiome in the evaluation of the gut microbiome’s impact on locomotion, we developed a novel gut microbiome analysis method. This method eliminated individual genetic backgrounds, allowing a more accurate evaluation of the gut microbiome’s influence on locomotion. The method compared the locomotion of mice with the same individual mice after remodeling their gut microbiome through fecal microbiota transplantation (FMT), thereby accurately identifying the intestinal microbes associated with locomotion. For the first time, we determined the effects of the gut microbiome on locomotion through comparing locomotion within the same mouse after remodeling the gut microbiome under the same environmental conditions.

## 2. Results

### 2.1. A Random Colonization of Human Gut Microbiome into Conventional Mice through FMT Had a Different Impact on Locomotion

To identify the specific bacteria about the regulation of locomotion, we utilize the concept of subgroup analysis, which helps uncover the cause of complex issues in large data sets, we performed in vivo experiment. The aim was to identify the intestinal bacteria that regulate locomotion through randomized subgroup analysis. We achieved this through feeding a fresh fecal sample to conventional mice (C57BL/6) whose gut microbiome had been depleted through a mixture of three broad-spectrum antibiotics and nystatin (Figure 1A).

The effect of the subset FMT with the human gut microbiome in 21 conventional mice (15 females and 6 males) was vastly different on locomotion (Figure 1B,C). In order to exclude genetic factors and evaluate the effect of the gut microbiome on locomotion change alone, the wire hanging records were monitored in the same mice after replacing their original gut microbiome with a human gut microbiome through the FMT. The changes in the locomotion of the mice over the three-month experimental period can be grouped into three categories: the group with increased locomotion, in which the hanging time on the wire increased by 26.6 ± 4.8 s (SL; strong locomotion); the group with unchanged locomotion, in which the hanging time on the wire remained within the range of 4.3 ± 1.8 s (ML; medium locomotion); and the group with decreased locomotion, in which the hanging time on the wire decreased by −32.6 ± 11.1 s (WL; weak locomotion) (Figure 1B,C). The SL group consisted of six females and one male, while the ML group consisted of four females and three males, and the WL group consisted of five females and two males. The changes of locomotion in each group were confirmed via the histological examination, which showed a high degree of muscle fiber accumulation in SL, medium accumulation in ML, and the lowest accumulation in WL (Figure 1D). The levels of blood glucose and lipid profiles did not change significantly before or after the gut microbiome replacement in the mice (Appendix A). These results suggest that a different subset of the human gut microbiome randomly replaced the original gut microbiome in each mouse, resulting in different effects on locomotion.

### 2.2. Different Types of Gut Microbiome Were Established in Each of the Experimental Mice after FMT with Human Feces

The differential effects on locomotion after gut microbiome replacement prompted us to compare the compositional changes of the gut microbiome before and after FMT with human feces in the mice (Appendix A). The identified 16S rRNA sequences, represented as operational taxonomic units (OTUs), were classified into nine different phyla: Bacteroidetes (49.265%), Firmicutes (35.596%), Verrucomicrobia (11.224%), Proteobacteria (1.973%), Actinobacteria (0.842%), Tenericutes (0.501%), Patescibacteria (0.357%), Cyanobacteria (0.24%), and Lentisphaerae (0.001%) (Appendix A and Appendix A). The OTUs were further classified to the species level.

The comparison of the OTUs showed a clear difference before and after the FMT in all of the mice, indicating that the replacement of their original gut microbiome with those of the humans was successful, as shown in Appendix A. The differences were most notable at the genus and species levels (Appendix A). In accordance with the individual variation in locomotion after the FMT, the composition of the replaced gut microbiome was very different from one mouse to another (Figure 2, Appendix A, and Appendix A).

Before the FMT, the main bacteria that constituted the original gut microbiomes at the phylum level were Bacteroidetes (52.614% in SL, 56.302% in ML, and 55.709% in WL), Firmicutes (44.348% in SL, 40.159% in ML, and 41.932% in WL), and Patescibacteria (2.483% in SL, 2.917% in ML, and 1.831% in WL); at the class level, *Bacteroidia* (52.614% in SL, 56.302% in ML, and 55.709% in WL), *Clostridia* (39.625% in SL, 36.717% in ML, and 40.097% in WL), and *Bacilli* (4.24% in SL) or *Saccharimonadia* (2.917% in ML, and 1.831% in WL); at the order level, *Bacteroidales* (52.614% in SL, 56.301% in ML, and 55.709% in WL), *Clostridiales* (39.626% in SL, 36.717% in ML, and 40.097% in WL), and *Lactobacillales* (4.239% in SL) or *Saccharimonadales* (2.917% in ML, and 1.831% in WL); and at the family level, *Muribaculaceae* (50.047% in SL, 54.492% in ML, and 51.805% in WL), *Lachnospiraceae* (26.267% in SL, 23.014% in ML, and 27.684% in WL), and *Ruminococcaceae* (9.911% in SL, 11.034% in ML, and 9.571% in WL). However, the composition of bacteria changed three months after the FMT: at the phylum level, Bacteroidetes (47.46% in SL, 49.487% in ML, and 50.848% in WL), Firmicutes (41.797% in SL, 34.258% in ML, and 30.733% in WL), and Verrucomicrobia (7.853% in SL, 12.561% in ML, and 13.258% in WL); at the class level, *Bacteroidia* (47.46% in SL, 49.487% in ML, and 50.849% in WL), *Clostridia* (36.701% in SL, 29.373% in ML, and 22.466% in WL), and *Verrucomicrobiae* (7.853% in SL, 12.561% in ML, and 13.259% in WL); at the order level, *Bacteroidales* (47.408% in SL, 49.432% in ML, and 50.84% in WL), *Clostridiales* (36.701% in SL, 29.373% in ML, and 22.466% in WL), and *Verrucomicrobiales* (7.853% in SL, 12.561% in ML, and 13.259% in WL); at a family level, *Muribaculaceae* (39.713% in SL, 36.466% in ML, and 41.233% in WL), *Lachnospiraceae* (21.051% in SL, 15.364% in ML, and 11.465% in WL), and *Ruminococcaceae* (14.291% in SL and 12.639% in ML) or *Akkermansiaceae* (13.263% in WL) (Figure 2 and Appendix A). These results indicate that the human gut microbiome is not only more diverse, but also its composition significantly differs from that of mice.

### 2.3. The Differences in Locomotion Were Correlated with the Shifts in the Composition of the Gut Microbiome

After confirming the individual differences in the composition of the replaced gut microbiome and its correlation with locomotion, the gut microbiome of each mouse was analyzed. The α-diversity metrics, which measure both richness and evenness, showed that different subsets of the human gut microbiome were replaced in each group of mice (Figure 3 and Appendix A, and Appendix A). Although the α-diversity indices considering both richness (Fisher’s alpha) and evenness (Shannon) showed slight differences between the three groups, separate measurements of richness and evenness visualized the structural differences in the ecological community. The evenness indices before and after the FMT were similar, but the richness indices were increased after the FMT, suggesting that the human gut microbiome is more diverse, although the compositional characteristics of the human and mouse gut microbiomes are similar. Furthermore, the richness and evenness diversity indices of the mice with stronger locomotion (SL) were higher than in the other two groups, indicating a more diverse gut microbiome in SL.

Other than α-diversity analyses, β-diversity analyses have also confirmed that the replaced gut microbiome is much more diverse than its original gut microbiome in each mouse. As shown in Figure 4A,B, both β-diversity metrics, as measured via NMDS and MDS plots, show that the features are more diverse after gut microbiome replacement.

To examine the impact of gut microbiome on locomotion, we compared the gut microbiome composition before the FMT (T0) and three months after the FMT (T3). The comparison using unsupervised hierarchical clustering of the most abundant Operational Taxonomic Units (OTUs) based on the Bray–Curtis distance revealed that the replacement of the gut microbiome affected the three groups of mice differently. The unsupervised hierarchical cluster analysis at T0 showed that each group of mice was completely distinct from each other (as seen in Figure 4C, left). However, when considering locomotion differences, individual mice from the three groups clustered together (as seen in Figure 4C, right). This study only took into account locomotion changes within individual mice, so the gut microbiome at the starting point did not show any group differences because the starting point was not related. However, the gut microbiome’s impact on locomotion is evident in the unsupervised hierarchical cluster analysis at T3, which shows the gut microbiome composition grouping individual mice according to differences in locomotion.

### 2.4. Different Microbial Communities Were Established in Each of the Three Groups of Mice following the FMT

The differences between the gut microbiomes of the three groups of mice (SL, ML, and WL) were analyzed through constructing phylogenetic trees. A maximum-likelihood phylogeny of each microbiome was built based on the 16S rDNA sequence (Figure 5). The diversity of the gut microbiomes of all three groups expanded after the replacement of the gut microbiome: in the SL group, there was a 60% increase from 150 species before FMT to 240 species after FMT, while in the ML group, there was a 54.8% increase from 155 species to 240 species, and in the WL group, an increase of 55.7% from 149 species to 232 species (Figure 5). In addition to the increased diversity, the composition of the intestinal bacteria in the gut microbiome changed dramatically after the replacement.

The phylogenetic analysis showed that the replaced gut microbiomes of SL, ML, and WL were different, despite the fact that their original gut microbiomes were not different from each other (Figure 5). Consistent with the phylogenetic analysis, the co-occurrence network analysis also demonstrated that the microbial communities became more diverse after the gut microbiome was replaced in all three classified groups (Figure 6 and Appendix A). The 27 communities in SL before the gut microbiome replacement expanded to 56, while the 27 communities in ML became 65 and the 38 communities in WL became 45. There was no significant difference in the number of nodes and edges within the microbial communities, which suggests that bacteria interact with each other in a similar manner. The higher number of microbial communities in SL compared to ML and WL indicates that locomotion is associated with a more diverse gut microbiome.

### 2.5. Intestinal Microbes That Affect Locomotion Were Identified at the Species Level

The above gross gut microbiome analyses revealed a clear correlation between gut microbiome composition and locomotion. However, the group analyses did not visualize a specific group of bacteria responsible for either promoting or reducing locomotion (Figure 2, Figure 3, Figure 4, Figure 5 and Figure 6). Due to the limitations of group analysis in identifying intestinal microbes at the species level, we utilized the concept of fold change at the log2 scale and a linear correlation to analyze the abundance of intestinal microbes in relation to locomotion (Figure 7). Among the bacterial species, *Lacrimispora indolis*, *Pseudoflavonifractor phocaeenis*, and *Alistipes senegalensis* were most abundant in SL, indicating a positive correlation with locomotion. On the other hand, *Sphingobacterium cibi*, *Prevotellamassilia timonensis*, *Parasutterella excrementihominis*, *Faecalibaculum rodentium,* and *Muribaculum intestinale* were most abundant in WL, indicating a negative correlation with locomotion. The bacteria associated with strong locomotion were classified under the phylum Firmicutes and Bacteroidetes, while those associated with weak locomotion belonged to the phylum Bacteroidetes, Proteobacteria, and Firmicutes.

## 3. Discussion

It has been well established that the gut microbiome plays a crucial role in energy extraction from food within the host through fermentation processes and an increase in villous vascularization [34,35]. However, the role of the gut microbiome is not limited to just energy extraction. Recent research has shown that the gut microbiome has a significant impact on a wide range of physiological processes, including metabolism, digestion, immunity, and brain function [2,3,4,5,6]. Additionally, the gut microbiome serves as a crucial epigenetic factor that modulates the expression of our own genes [36,37,38,39,40].

Considering the significant role of the gut microbiome, one would expect it to play a critical role in determining its host’s locomotion. However, previous research has not shown a clear association between the gut microbiome and locomotion. Although no previous paper has shown a clear link between the gut microbiome and locomotion, various studies have predicted a potential link between the gut microbiome and locomotion [25,26,27,28]. The gut microbiome is a complex ecosystem composed of tens of trillions of bacteria and fungi, with hundreds of different species and thousands of strains [41]. The number of microorganisms in the gut microbiome can vary greatly between individuals. Given the diversity of an individual’s gut microbiome and the vast number of intestinal microbes, the composition of the gut microbiome could consist of a variety of combinations of intestinal microbes, meaning that some gut microbiomes could negatively impact locomotion while others could positively impact it [42,43]. As a result, it is natural to see a mix of evidence in the association between locomotion and the gut microbiome in research.

The comparison of the gut microbiome between control and experimental groups through metagenomics has some limitations in its analysis. To address these limitations, we transferred human fecal microbiomes into mice and observed changes in phenotype in the same individual mice. The use of subset analysis within a large dataset is very useful for uncovering differences that might otherwise go unnoticed. Therefore, the transfer of the human gut microbiome (large dataset) to mice and subsequent analysis of the transplanted microbiome in mice (subset data) offers a valuable approach for analyzing the human gut microbiome. Additionally, through comparing changes in phenotype, such as changes in locomotion, within the same individual mouse, genetic factors affecting locomotion can be controlled for, thus revealing the sole impact of the gut microbiome on phenotypes [33].

Our experimental approach has revealed that the gut microbiome has a dual impact on locomotion: some gut microbiomes positively affect locomotion, while others negatively affect locomotion (Figure 1). Alpha and beta diversity analyses confirmed that these gut microbiomes differ in terms of the composition and diversity of their constituent intestinal microbes (as shown in Figure 2, Figure 3, Figure 4 and Figure 5). These findings were further supported by the co-occurrence network analysis, which indicated that the gut microbiome affecting locomotion differed from others (Figure 6). Our findings demonstrate that the gut microbiome contributes positively or negatively to locomotion.

Our FMT-based gut microbiome analysis not only revealed the potential existence of a positive or negative relationship between the gut microbiome and locomotion, but also identified the specific gut microbiomes that were positively or negatively associated with locomotion (Figure 7). As depicted in Figure 7, *Lacrimispora indolis*, *Pseudoflavonifractor phocaeensis*, and *Alistipes senegalensis* had a positive effect on locomotion, while *Sphingobacterium cibi*, *Prevotellamassilia timonensis*, *Parasutterella excrementihominis*, *Faecalibaculum rodentium*, and *Muribaculum intestinale* had a negative impact. *L. indolis* is a bacterium commonly found in soils and the feces of birds and mammals and is reported to have the metabolic potential to utilize a wide assortment of fermentable carbohydrates and intermediates including citrate, lactate, malate, succinate, and aromatics [44]. *P. phocaeensis* is an anaerobic, Gram-negative bacterium [45]. Although *Alistipes senegalensis* is known to contribute primarily to disease and, conversely, to serve as a healthy phenotype, *Alistipes senegalensis* has been isolated from the fecal flora of asymptomatic patients [46,47]. *Sphingobacterium cibi* was isolated from a food-waste compost sample collected in Taichung, Taiwan [48]. *P. timonensis* was isolated from the stool specimen of a patient hospitalized for the treatment of a melanoma [49]. *P. excrementihominis*, a strictly anaerobic, non-spore-forming, Gram-negative coccobacillus, was isolated from human faeces [50]. *F. rodentium* was isolated from the faeces of a laboratory mouse and found to produce lactic acid as a major metabolic end product [51]. *M. intestinale* is strictly anaerobic and can degrade the galactose to form β-D-glucose 6-phosphate [52]. Overall, while the specific mechanisms underlying the relationship between the gut microbiome and locomotion are still not fully understood, there is growing evidence to suggest that the gut microbiome can influence host motor function, coordination, and activity levels. Further research is needed to identify specific gut microbes and mechanisms underlying these effects, and to determine the potential therapeutic applications for individuals with neurological and motor disorders.

Finally, this study has successfully demonstrated that the in vivo subset analysis of the human gut microbiome using an animal model is a valuable approach to study this complex and heterogeneous population of trillions of microbes. We believe that this concept could be applied to identify the gut microbes associated with other human phenotypes or diseases.

## 4. Materials and Methods

### 4.1. Study Design and Animal Experiment

Twenty-one C57BL/6 mice (15 females and 6 males) purchased from Animal Facility of Aging Science in Korea Basic Science Institute (Gwangju, Republic of Korea) were used in this study. The mice were housed individually in a sterile environment and had access to sterilized food and water. They were maintained under a specific light/dark cycle and temperature. After a week of acclimation, the mice were weighed, and their feces and blood were collected. To deplete their endogenous microbiota, the mice were given water containing antibiotics (1 g/L ampicillin, 0.5 g/L kanamycin, and 0.5 g/L cefoxitin; Sigma-Aldrich, St. Louis, MO, USA) and an antifungal (0.5 g/L nystatin) for one week.

Optimized medium was prepared through mixing 31 g of 19 edible plants, including oatmeal, brown glutinous rice, spinach, mung bean, adlay, ginger, broccoli, bean sprout, philmuri sprout, cham namul, beetroot, hulled hempseed, dried orange peel, ginseng, licorice, morus bark, buckwheat, balloon flower, and green onion leaves; 2.4 g of beef extract (BD); 10 g of Bacto-peptone (BD); 4.0 g of Na_2_HPO_4_; 5.0 g of yeast extract (Duchefa biochemie); 1.5 g of glucose (Duchfa biochemie); 0.5 g of soluble starch; 1 g of L-cysteine-HCl (Duchefa biochemie); 2.0 g of trypticase peptone (BD); 1 g of NaCl (Duchefa biochemie); 0.2 g of arginine (Daejung); 0.2 g of sodium pyruvate; 1 g of urea; 0.2 g of CaCO_3_ (Duchefa biochemie); 2.0 g of Polypeptone (BD); 0.001 mg of Hemin; 2 mL glycerol (Duchefa biochemie); 1 mL mineral solution (composed of 0.72 mg/L MnCl_2_ × 4 H_2_O, 0.2 g/L FeSO_4_ × 7 H_2_O, 0.2 g/L CoCl_2_ × 6 H_2_O, 0.2 g/L ZnSO_4_ × 7 H_2_O, 0.02 g/L Na_2_MoO_4_ × 2 H_2_O, 0.1 g/L NiCl_2_, 0.02 g/L CuSO_4_ × 5 H_2_O, 0.02 g/L H_3_BO_3_, 0.6 g/L MgSO_4_. 7 H_2_O, 0.6 g/L CaCl_2_. 2 H_2_O, and 0.1 g/L (NH_4_)_2_SO_4_ in double distilled water (dd H_2_O)); 578 μL SCFA solution (a mixture of 15 mL acetic acid, 6 mL propionic acid, 1 mL n-valeric acid, 1 mL iso-valeric acid, 5 mL butyric acid, and 1 mL iso-butyric acid) in 948 mL of dd H_2_O. The medium was then autoclaved for 22 min at 121 °C, cooled to below 40 °C, and supplemented with 2 mL of filter sterilized (0.22 μm) vitamin solution (prepared through mixing 0.02 g of biotin, 0.02 g of cyanocobalamin, 0.03 g of P-amino benzoic acid, 0.05 g of folic acid, 0.15 g of pyridoxine HCl, 0.05 g of thiamine HCl, 0.05 g of riboflavin, 0.05 of L-ascorbic acid, 0.02 g of coenzyme q 10, 0.2 g of L-glutamine, 0.03 g of dimethylglycine, 0.03 g of inositol, 0.03 g of niacin, 0.03 g of L-carnitine, 0.03 g of methylsulfonylmethane, and 100 μL of vitamin K1 in 1 L of dd H_2_O) and 50 mL of pig blood. The medium was deoxygenated and stored in an anaerobic container before use [53].

Fecal samples from 10 healthy volunteers were collected and mixed in the optimized medium to make mixed feces containing various human gut microbes. The fecal medium was created on the scheduled day for oral gavage and stored at room temperature in anaerobic containers. The mixed human feces medium was used to perform FMT in mice. The FMT was performed twice a week for a total of three months through administering 20 μL of the mixture to the mice via oral gavage. All of the mice were weighed after finishing feeding of the human fecal sample, and the feces and blood of the mice were individually collected.

### 4.2. Wire Hanging Test

This test was used to assess forelimb strength. The apparatus consisted of a stainless-steel wire (50 cm length, 2 mm in diameter), fixed horizontally between two vertical supports and 60 cm above the bedding. The mice were forced to grasp the central position of the wire with their forepaws. The latency to fall from the wire to the bedding was measured. The trial was conducted three times for each mouse, and the average was the value used for evaluation. The resting pause between consecutive attempts was 3 min.

### 4.3. Analyses of Biochemical Parameters

The serum levels of total cholesterol (TCHO), triglyceride (TG), and high-density lipoprotein cholesterol (HDL-CHO) were determined through enzymatic methods using commercial assay kits (Asan Pharmaceutical Co., Seoul, Republic of Korea) as described previously [54,55]. In brief, the low-density lipoprotein cholesterol (LDL-CHO) levels were calculated using Friedewald’s equation [(LDL-CHO) = (TCHO) − ((HDL-CHO) − (TG)/5)].

The blood glucose level was determined using glucose test strips and a handheld blood glucose meter (Accu-Chek Active; Roche Diagnostic GmbH, Mannheim, Germany).

### 4.4. Histological Analysis

The histological analysis was determined using a method described previously [56]. Briefly, muscle tissues were prepared with mice at 3 months of the experimental intervention and were used for microscopic analysis after being stained with H&E. Immediately after isolation, muscle tissue sections were fixed in 10% neutral buffered formalin and were paraffin-embedded. Serial 6 µm thick sections were randomly selected. Paraffin was removed from the tissue sections with hot water. The tissue sections were placed on microscopic slides, and the slides were air-dried and baked overnight at 65 °C. Finally, the tissue sections were stained with hematoxylin and eosin (Vector laboratories Inc., Newark, CA, USA) according to standard laboratory procedure. The H&E-stained tissue sections were observed under a light microscope (AmScope, T690C-PL, Irvine, CA, USA), and images were captured with a microscopic digital camera (AmScope, MU-1803, Irvine, CA, USA).

### 4.5. DNA Extraction and 16S rRNA Gene Sequencing

Total bacterial genomic DNA from each sample was extracted using the phenol-chloroform isoamyl alcohol extraction method as described previously [57,58]. Briefly, samples suspended in lysis buffer (200 mM NaCl, 200 mM Tris-HCl (pH 8.0), 20 mM EDTA) were processed via bead-beating. Genomic DNA was recovered from the aqueous phase using Phenol:Chloroform:Isoamylalcohol. DNA was precipitated with the addition of 3 M sodium acetate followed by isopropanol. After rinsing with 70% ethanol and drying, the DNA pellet was dissolved in TE buffer (10 mM Tris-HCl (pH 8.0), 1 mM EDTA). The concentration and purity of the extracted DNA were measured using a BioSpec-nano spectrophotometer (Shimadzu Biotech, Kyoto, Japan), and the integrity was evaluated on a 1% (*w*/*v*) agarose gel.

Metagenome sequencing analyses of the gut microbiome DNA samples were processed and sequenced by a commercial company, ebiogen, Inc. in Republic of Korea. Briefly, each sequenced sample was prepared according to the Illumina 16S Metagenomic Sequencing Library protocols, and the genes were amplified using 16S V3–V4 primers; 16S Amplicon PCR Forward Primer with a sequence of 5′-TCGTCGGCAGCGTCAGATGTGTATAAGAGACAGCCTACGGGNGGCWGCAG-3′ and 16S Amplicon PCR Reverse Primer with a sequence of 5′-GTCTCGTGGGCTCGGAGATGTGTATAAGAGACAGGACTACHVGGGTATCTAATCC-3′. Input genomic DNA was amplified with 16S V3–V4 primers. A subsequent limited-cycle amplification step was performed to add multiplexing indices and Illumina sequencing adapters. The final products were normalized and pooled using PicoGreen, and the size of the libraries was verified using the Agilent TapeStation DNA ScreenTape D1000 system (Agilent Technologies, Santa Clara, CA, USA). Finally, the pooled libraries were sequenced (2 × 300) using the MiSeq platform (Illumina, San Diego, CA, USA). The amplicon error was modelled from merged fastq using DADA2 (ver.1.10.1); noise sequence was filtered out, errors in marginal sequences were corrected, chimeric sequences and singletons were removed, and sequences were de-replicated [59]. The raw data were deposited in the repository at figshare (https://doi.org/10.6084/m9.figshare.22567054.v1) accessed on 6 April 2023.

### 4.6. Data and Statistical Analyses

All data and statistical analyses were determined as described previously [57,58]. In brief, the Q2-Feature classifier is a Naive Bayes classifier trained based on the SILVA v138 reference (region V3–V4) database (https://www.arb-silva.de/) accessed on 20 July 2021 to classify bacterial species. We used the program to classify our datasets after setting De-noise-single function as the default parameter. The q2-diversity under the option of “sampling-depth” was used for the diversity calculation and statistical tests. We employed a sequencing quality score threshold of at least 20 and rarefaction depth at 11,510. After confirming the quality of sequencing results, the sequencing results in “table.qzv” files were filtered using the threshold values in QIIME 2. The metagenomic data OTU and taxonomic classification tables were imported into the phyloseq (1.28.0) package in R version 3.6.1 for visualization of alpha and beta diversity. Statistical analysis was performed using Kruskal–Wallis rank sum test for alpha diversity. To detect statistical differences in beta diversity metrics between groups, we used permutational multivariate analysis of variance (PERMANOVA) in the vegan package in R. ADONIS was used with 999 permutations in the vegan package in R to quantify the effect size of variables explaining Bray–Curtis distance. All *p*-value was corrected with Benjamini and Hochberg’s adjustment, and significance was declared at *p* < 0.05.

### 4.7. The α-Diversity Analysis for Relative Abundance Evaluation of Material and Microbiome

The α-diversity analysis for relative abundance was determined as described previously [57,58]. We used the phyloseq (1.28.0) and metagenomeSeq (1.16.0) packages to identify the central taxa present in each group. The metadata, OTUs, and taxonomic classification tables were imported into the phyloseq package, and the data were processed as instructed [60,61]. The phyloseq class object was converted to metagenomeseq objects and was normalized via cumulative-sum-scaling (CSS), which was specially built for metagenome data in the bioConductor package metagenomeSeq (1.16.0). Normalized data were converted to phyloseq class objects in R for further analysis and visualization.

Normalized OTU data were used for abundance calculation, and each taxonomic level was glommed for plotting. For clear visualization of abundance data, taxa were collected into “other” if they had relative abundances below 5% except at the phylum and class levels.

### 4.8. The β-Analysis for Relative Abundance Evaluation of Material and Microbiome

The β-diversity was computed for non-metric multidimensional scaling (NMDS) from log-transformed OTU data using Bray–Curtis dissimilarity in the vegan package as described previously [57,58]. Using the metaMDS function in the ‘vegan’ package, NMDS was performed on the Bray–Curtis dissimilarity matrix, which reduced dimensionality while retaining as much information as possible about relationships among samples.

### 4.9. Construction of Heatmap and Phylogenetic Tree

A heatmap and cluster analysis were generated using the relative abundances of genera from all OTU values or core abundant OTU values in the Heatplus (2.30.0) package from Bioconductor and the vegan package in R as described previously [33,62]. Average linkage hierarchical clustering and Bray–Curtis distance metrics were used for cluster analysis and heatmap generation, respectively. Unsupervised prevalence filtering was performed with a 5% threshold in total samples to collect the most abundant taxa for heatmap generation.

Phylogenetic trees for each sampling site were constructed from row sequences without any filtering to show direct visualization of sample richness with relation to taxonomic classification as described previously [33,62]. Briefly, taxa that could not be classified down to the species level were reclassified based on the NCBI accession number using the taxonomizr (0.5.3) package in R. Then, 16S rRNA sequences from each sampling site were aligned in ClustalW with a default parameter, and the resulting alignments were used to construct the maximum-likelihood phylogenetic trees in MEGAX with 500 bootstrap replicates. All phylogenetic trees were visualized in iTOL.

### 4.10. Co-Occurrence Network Construction

Co-abundance networks were created using a permutation-renormalization-bootstrap network construction strategy as described previously to observe the microbial co-occurrence relationships through the mice’s rotarod record change [33,62]. Briefly, non-normalized abundance data was uploaded to CoNet, a Java Cytoscape plug-in. All networks were in-dependently constructed through splitting the OTU abundance matrix into High, Medium, and Low groups. The microbial networks and links or edges were obtained from OTU occurrence data. Multiple ensemble correlation methods in CoNet were used to identify significant co-presences across the samples while OTUs that occurred in less than three samples were discarded (“row_minocc” = 3). Five similarity measures, including Spearman and Pearson correlation coefficients, the Mutual information Score, and the Bray–Curtis and Kullback–Leibler Dissimilarity, were calculated using CoNet for the creation of an ensemble network, and the *p*-value was merged using Brown’s method. The *p*-value was corrected using the Benjamini–Hochberg correction method (adjusted *p*-value < 0.05). If at least two of the five metrics suggested significant co-abundance between the two OTUs, the relationship was kept in the final network to be represented as an edge. The final co-occurrence network model was displayed via the igraph package in R by using an implementation of the Louvain algorithm to identify communities within each network so that the modularity score of each OTU was maximized within a given network.

### 4.11. Differential Abundance

DESeq2 (version 1.24.0) was used to estimate the fold-change of taxa in the gut microbiome according to the wire suspension hanging time change groups [63]. Taxa that were not observed in at least 0.5% of samples were excluded from the DESeq2 analyses.

### 4.12. Quantification and Statistical Analysis

The differences in the relative abundance of bacterial populations containing feces were analyzed using the Mann–Whitney sum rank tests in R software (R 4.2.0, RStudio, PBC, Boston, MA, USA). Significance was declared at *p* < 0.05 with Benjamini and Hochberg’s adjustment. Graphs were prepared using R software and ImageGP [64].

### 4.13. Ethics Approval

All the experimental procedures complied with the ARRIVE guidelines, and the Institutional Animal Care and Use Committee of the Korea Basic Science Institute approved the animal protocols (KBSI-IACUC-23-12).

## Figures and Tables

**Figure 1 ijms-24-11392-f001:**
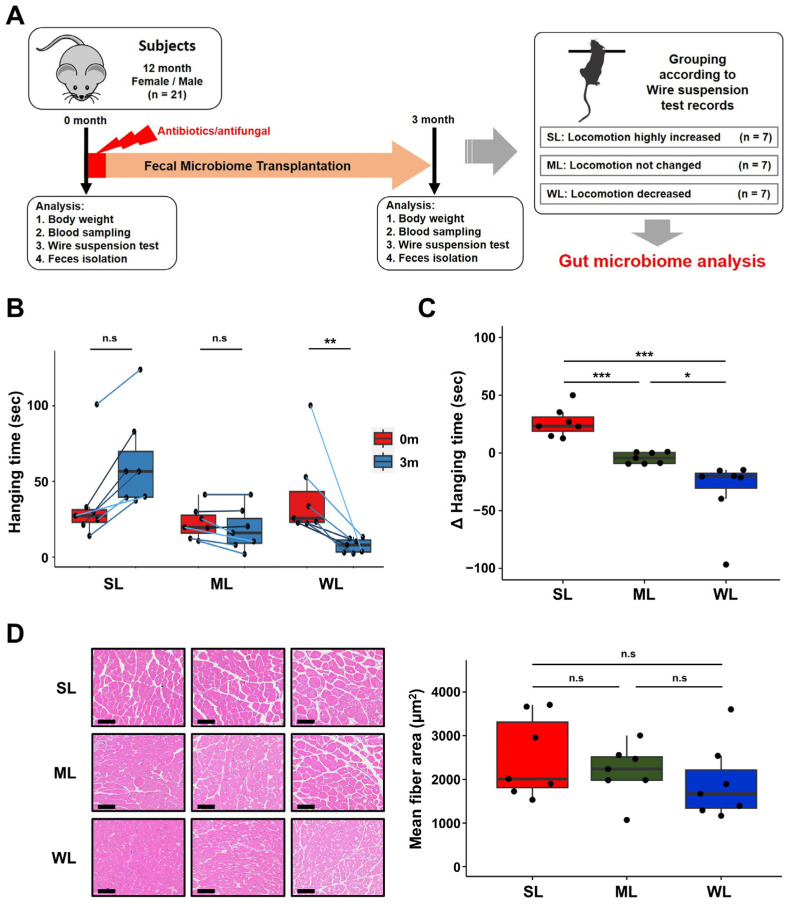
Grouping of mice based on changes in locomotion before and after FMT. (**A**) A schematic diagram of the experimental design. (**B**) The comparison of wire hanging performance of the experimental mice before and after FMT. (**C**) Changes in locomotion of the experimental mice after FMT. (**D**) Representative histological images of H&E-stained muscle tissue (scale bar = 100 μm) from the mice with light, medium, and heavy locomotion (SL, ML, and WL, respectively). The values in the figure are expressed as the mean ± standard deviation (SD) and the level of significance is indicated by asterisks (*). * *p* < 0.05, ** *p* < 0.01; *** *p* < 0.001. “n.s.” stands for “not significant” and indicates a *p*-value greater than 0.05.

**Figure 2 ijms-24-11392-f002:**
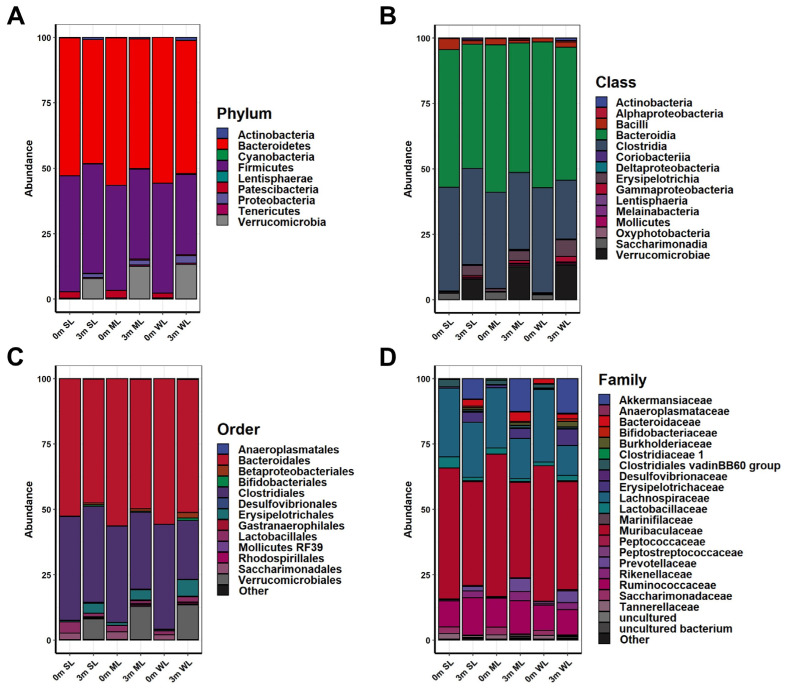
The composition of the gut microbiome in experimental mice before and after FMT. The relative changes in the gut microbiome composition were analyzed at the (**A**) phylum, (**B**) class, (**C**) order, and (**D**) family levels. The average abundance values of each group were calculated to obtain the abundance values.

**Figure 3 ijms-24-11392-f003:**
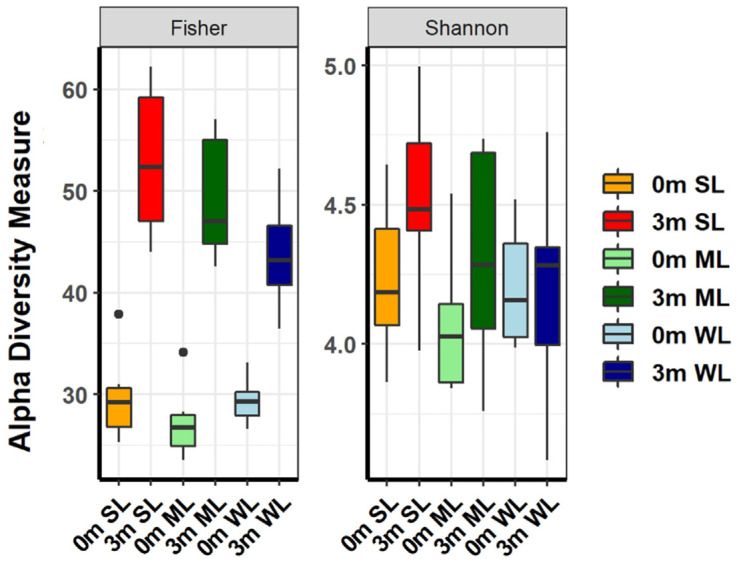
The α-diversity indices of the gut microbiome of the SL, ML, and WL groups. The species richness and diversity, as calculated using Fisher’s alpha and Shannon diversity tests, are shown for before (0 m) and after (3 m) FMT for each of the SL, ML, and WL groups.

**Figure 4 ijms-24-11392-f004:**
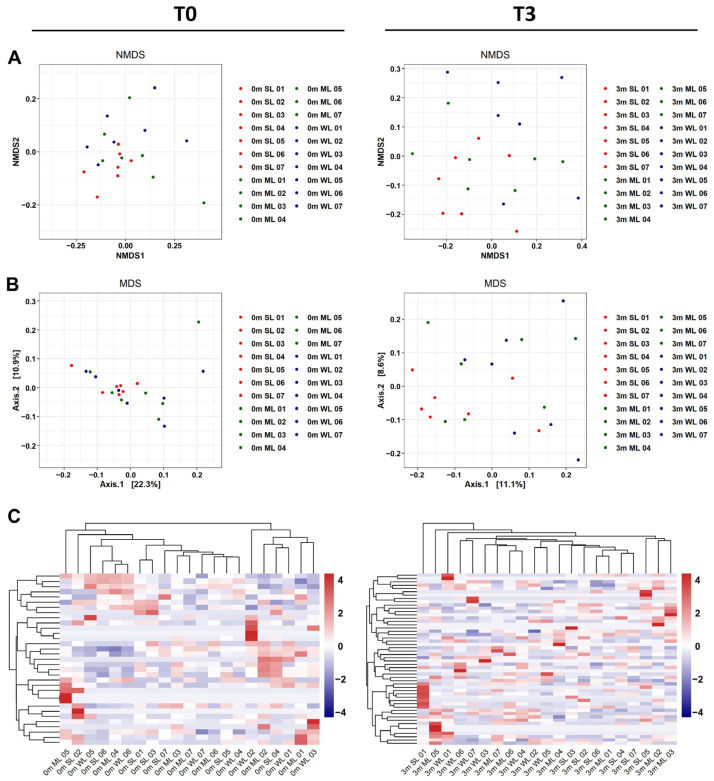
The β-diversity comparison of the gut microbiome of the SL, ML, and WL groups. (**A**) Non-metric multidimensional scaling (NMDS) plots are shown, depicting the differences in the gut microbiome before (T0, left) and after (T3, right) FMT in the SL (01~07 sample), ML (01~07 sample), and WL (01~07 sample) groups, based on Bray–Curtis distances calculated using operational taxonomic units (OTUs). (**B**) Multidimensional scaling (MDS) plots are shown, illustrating the differences in the gut microbiome before (T0, left) and after (T3, right) FMT in the SL (01~07 sample), ML (01~07 sample), and WL (01~07 sample) groups, based on Bray–Curtis distances calculated using OTUs. (**C**) Heatmaps of the microbial composition of the SL (01~07 sample), ML (01~07 sample), and WL (01~07 sample) groups before (T0, left) and after (T3, right) FMT are shown, based on the Bray–Curtis distance matrix measured at the phylum level.

**Figure 5 ijms-24-11392-f005:**
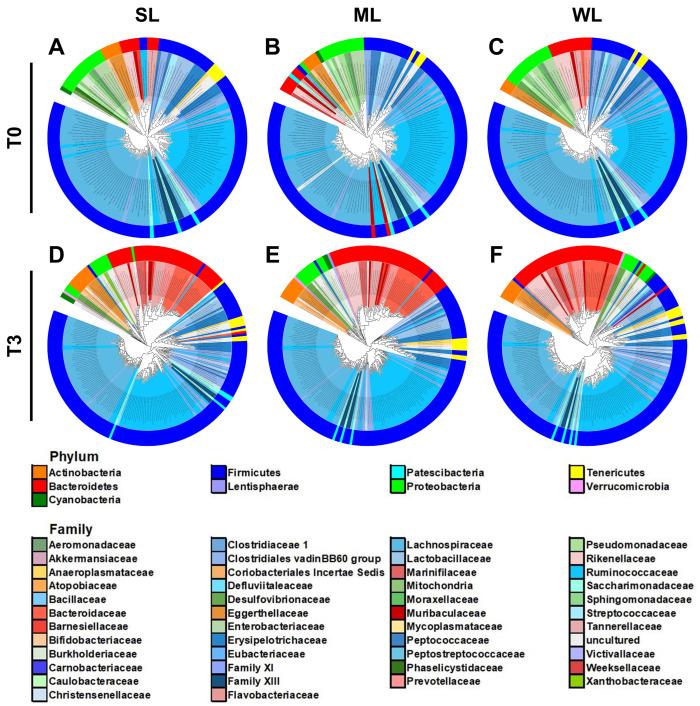
The maximum-likelihood phylogenetic tree comprising of the gut microbiome taxa in the SL, ML, and WL groups. The outer rings of the circular dendrogram represent the phylum level, while the inner layer represents the family level. (**A**–**C**) The SL, ML, and WL groups, respectively, before (T0) FMT. (**D**–**F**) The SL, ML, and WL groups, respectively, after (T3) FMT.

**Figure 6 ijms-24-11392-f006:**
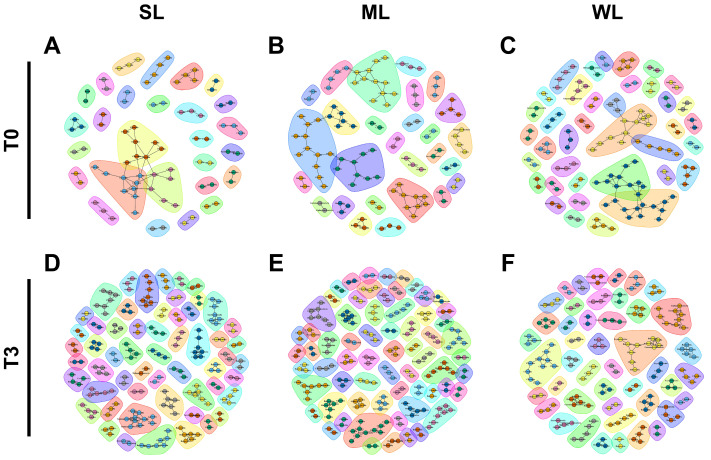
The co-occurrence network analysis using the ReBoot algorithm for the SL, ML, and WL groups. The color-coded network graphs indicate the co-occurring and mutual exclusion interactions between operational taxonomic units (OTUs). Black letters in the nodes correspond to the class level of the OTUs. Transparent shapes represent network communities determined using the Louvain modularity algorithm. (**A**–**C**) depict the SL, ML, and WL groups, respectively, before (T0) FMT. (**D**–**F**) show the SL, ML, and WL groups, respectively, after (T3) FMT.

**Figure 7 ijms-24-11392-f007:**
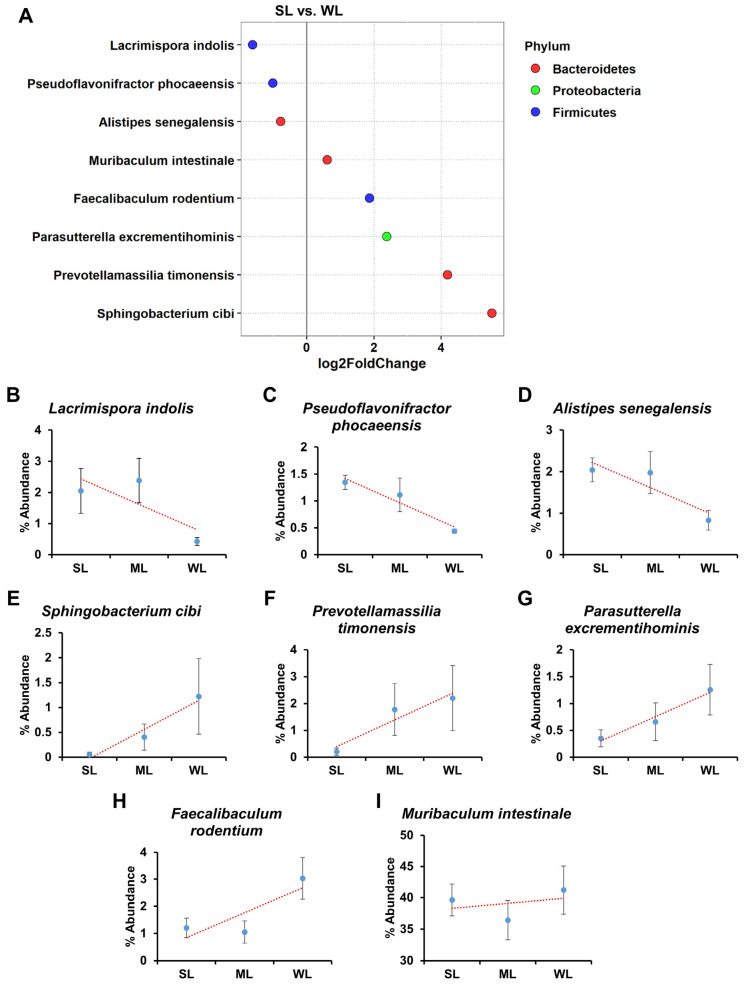
The key taxa changes between SL and WL according to differential abundance analysis. (**A**) The log2-fold change in abundance of the most abundant species in the gut microbiome of the SL and WL groups was analyzed using DESeq2 differential abundance analysis. Each point represents a comparison of species between the two experimental groups. Dot colors mean the phylum type. (**B**–**I**) The normalized abundances of eight significantly different bacterial species of interest that were identified from the differential abundance analyses are shown. Boxplots represent the normalized abundances in each group. The blue dot means the average of each group. The red line means the trend line between groups. *p*-value < 0.05 was considered as significant.

## Data Availability

The raw data were deposited in the repository at figshare (https://doi.org/10.6084/m9.figshare.22567054.v1) accessed on 6 April 2023.

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
