# Peer review of "Identification of the Intestinal Microbes Associated with Locomotion"

_ijms, 2023, doi:10.3390/ijms241411392_

Round 1
Reviewer 1 Report
Ahn et al. identified a number of locomotion-related gut bacteria by reshaping the original gut microbiome of mice by using human fecal microbiota transplantation (FMT) and comparing changes in locomotion in the same mice before and three after FMT. This paper on the topic of the relationship between locomotion capacity and gut microbiome is very interesting. Although some locomotion-related gut bacteria were found, a few points need to be clarified in the manuscript. Here are some comments on this paper and questions that I would like to discuss with the authors:
1. Lines 17-18 “We found that FMT affected locomotion in three different ways: positive, none, and negative.” It is obvious that there are only three possible outcomes of FMT affecting locomotion. It is better to present the results for positive, none the same, and negative separately rather than writing the above sentence.
2. There were 8 bacterial names in the keywords. It is better to delete these.
3. Could authors clarify the sex and number of the mice used in the experiment?
4. In Figure 1 B, could authors add points and connecting lines, which would allow readers to more clearly compare the changes in the locomotion ability of mice before and after FMT.
5. In Figure 1 B, ML group mice had lower hanging performances than the other two groups before FMT and remained lower after FMT. Does this indicate that the mice themselves, rather than the gut microbiome, determine their locomotion?
6. The mice were treated with a cocktail of antibiotics for 7 days before FMT, and as seen in Figure 1 A the month 0-time point was having been treated with antibiotics. However, through Figures 2 and 3 the mice’s gut microbiome species and diversity were not affected by antibiotics before the FMT. Can the authors give any explanation?
7. Lines 363-365 “After completing FMT with the new human microbiome, the mice were kept in a controlled laboratory environment for an additional three months in the above normal mouse facility for 3 months” It appears that the mice were kept for an additional 3 months after 3 months of FMT, so it was a 6 months experiment. However, it looks like the mice were sacrificed at 3 months.
8. Lines 128-134, sequencing methods were already written in the method section.
9. Mice were at least 15 months old after 3 months of FMT, and although they were not yet aged mice, 15-month-old mice already exhibited aging changes. In my opinion, it is necessary to set the non-FMT control group.
10. Lines 296-297, I don’t feel that this paper invented a new method for the gut microbiome.
11. Line 355, 10 healthy volunteers’ fecal samples were mixed in the optimized medium. Could authors give more information about the optimized medium?
12. Line 357, how long was the fecal medium stored at room temperature? How was the fecal medium preserved during the 3-month FMT?
13. There are some spelling mistakes, for example, line 211 Clostridia requires italics, line 360 ul should be μL, line 389 65 ℃ should be 65℃, line 425 SILVA needs version number, line 442 “Analysis for Relative Abundance” should be “analysis for relative abundance”.
Some spelling mistakes need to be corrected.
Author Response
Reviewer 1
Major revisions
- Lines 17-18 “We found that FMT affected locomotion in three different ways: positive, none, and negative.” It is obvious that there are only three possible outcomes of FMT affecting locomotion. It is better to present the results for positive, none the same, and negative separately rather than writing the above sentence.
Response to Comment:
As you recommended, we corrected none to none the same. Please refer to line 18. Thank you.
- There were 8 bacterial names in the keywords. It is better to delete these.
Response to Comment:
As you recommended, we removed the microbial name from the keywords. Please refer to line 29. Thank you.
- Could authors clarify the sex and number of the mice used in the experiment?
Response to Comment:
A total of 21 mice (15 females and 6 males) were used. There are 6 females and 1 male in the SL group, 4 females and 3 males in the ML group, 5 females and 2 males in the WL group. Please refer to line 90-91, line 101-103 and line 342. Thank you.
- In Figure 1 B, could authors add points and connecting lines, which would allow readers to more clearly compare the changes in the locomotion ability of mice before and after FMT.
Response to Comment:
As you suggested, we plotted the changes in locomotion ability of each mice before and after FMT for each group by adding dots and connecting lines. The graphs were added to the supplementary Figure S1. Thank you.
- In Figure 1 B, ML group mice had lower hanging performances than the other two groups before FMT and remained lower after FMT. Does this indicate that the mice themselves, rather than the gut microbiome, determine their locomotion?
Response to Comment:
Even if multiple mice are subjected to FMT using the same feces, the same intestinal microbes will not colonize. In other words, we predicted that all mice would have different gut microbiome after FMTs. We also measured and grouped changes in locomotion recordings of the same mouse to exclude characteristics of the mouse itself.
In Fig. 1B, low locomotion records of the ML group before FMT could be attributed to the low locomotion of the mice themselves, whereas similar locomotion records after FMT were because of either not forming communities of locomotion regulating microbes or colonizing balanced gut microbes of both increasing and decreasing locomotion. Thank you.
- The mice were treated with cocktail of antibiotics for 7 days before FMT, and as seen in Figure 1 A the month 0-time point was having been treated with antibiotics. However, through Figures 2 and 3 the mice’s gut microbiome species and diversity were not affected by antibiotics before the FMT. Can the authors give any explanation?
Response to Comment:
There seems to be a misunderstanding in the interpretation of Figure 1A. 0-month is before antibiotic/antifungal treatment. After all experiments were performed at 0 months, the antibiotic mixture was ingested for 7 days, and FMT was performed for 3 months. Therefore, in Figure 2 and Figure 3, 0-month is not unaffected by antibiotics, but a normal condition before treatment with antibiotics. Although data were not shown, it was confirmed that the species and diversity of intestinal microorganisms significantly decreased after antibiotic/antifungal treatment.
To avoid misinterpretation of Figure 1A, the antibiotics treatment indication has been moved to the right. Thank you.
- Lines 363-365 “After completing FMT with the new human microbiome, the mice were kept in a controlled laboratory environment for an additional three months in the above normal mouse facility for 3 months” It appears that the mice were kept for an additional 3 months after 3 months of FMT, so it was a 6 months experiment. However, it looks like the mice were sacrificed at 3 months.
Response to Comment:
It means "during the 3 months the FMT is in progress", not "for 3 months after completing the FMT". The paragraph was deleted because it was incorrectly expressed and redundant. Thank you.
- Lines 128-134, sequencing methods were already written in the method section.
Response to Comment:
As you suggested, the redundant ‘sequencing methods’ were deleted. Thank you.
- Mice were at least 15 months old after 3 months of FMT, and although they were not yet aged mice, 15-month-old mice already exhibited aging changes. In my opinion, it is necessary to set the non-FMT control group.
Response to Comment:
First of all, the subject of this thesis is not to look at changes due to aging, but to find human intestinal microbes that regulate movement. Second, individual characteristics due to aging would have been excluded through comparison before and after FMT of the same individual. Third, even if senescence was not excluded, senescence control would not have been considered because all mice experienced senescence equally. Thank you.
- Lines 296-297, I don’t feel that this paper invented a new method for the gut microbiome.
Response to Comment:
There are many studies using FMT, but the method of research on intestinal microbes, which compares the same organism to exclude individual characteristics, is the first method developed by our research team. Thank you.
- Line 355, 10 healthy volunteers’ fecal samples were mixed in the optimized medium. Could authors give more information about the optimized medium?
Response to Comment:
The optimized medium is a culture medium supporting universal growth of intestinal microbes. We recently submitted a paper about the optimized medium to Nature. This paper is in review state. According to your comment, we added citation in line 351 and reference [53]. Thank you.
- Line 357, how long was the fecal medium stored at room temperature? How was the fecal medium preserved during the 3-month FMT?
Response to Comment:
As specified in the manuscript, fecal medium was freshly prepared every scheduled day for oral gavage and stored for 2 hours at room temperature before use. Because it was made fresh every FMT day during the 3-month, the fecal medium was discarded without preservation after use. Thank you.
- There are some spelling mistakes, for example, line 211 Clostridia requires italics, line 360 ul should be μL, line 389 65 ℃ should be 65℃, line 425 SILVA needs version number, line 442 “Analysis for Relative Abundance” should be “analysis for relative abundance”.
Response to Comment:
As you noted, we have changed italics for all taxonomic ranks in line 129-132, line 140-163 and line 266-268. We also changed ‘ul’ to ‘μL’ in line 355, ‘65 ℃’ to ‘65℃’ in line 380, and ‘Analysis for Relative Abundance’ to ‘analysis for relative abundance’ in line 433. We noted the SILVA version number (v138) in line 416. Thank you.
Reviewer 2 Report
The manuscript presented by the authors provides a comprehensive study on the “Identification of the intestinal microbes associated with loco-motion”, using a large and valuable dataset. The study is important in deepening our understanding of “mothur with gut microbiota”. However, there are some issues for improvement in the manuscript.
Major reviews:
1. The figures need to be improved. Such as Figure 1B/D bar plot with error bar need change to boxplot with jitter dot. ImageGP (https://doi.org/10.1002/imt2.5) can generate high quality figures and with reproducible scripts.
2. The figure legend needs to detail descript the figures. Such as n=21 should be “n = 23” with blank before and after equal symbol; the sample size should be descripted for each group.
3. Figure 2 is a list of data. For the microbiome taxonomic level, just show the level of interest, such as phylum, genus, and focus the discussion around its differences.
4. Figure 3 is also listing all the alpha index. The finding should be the key to let reader to read and remember, not show all the data. Please only keep 1-2 panels, and remove them or as a supplementary figure.
5. Figure 4. Please show the legend in group, and in sample.
6. Figure 5. The comparing the tree? How to compare? Let the reads the read the detail in figure itself?
Author Response
Reviewer 2
- The figures need to be imporved. Such as Figure 1B/D bar plot with error bar need change to boxplot with jitter dot. ImageGP (https://doi.org/10.1002/imt2.5) can generate high quality figures and with reproducible scripts.
Response to Comment:
As you advised, we used ImageGP to replace the bar plots in Figures 1B-D with boxplots with jitter dots. Thank you.
- The figure legend needs to detail descript the figures. Such as n=21 should be “n = 23” with blank before and after equal symbol; the sample size should be descripted for each group.
Response to Comment:
We changed 'n=21' to 'n = 21'. The sample size is n = 7 for each group and was indicated in figure. Thank you.
- Figure 2 is a list of data. For the microbiome taxonomic level, just show the level of interest, such as phylum, genus, and focus the discussion around its differences.
Response to Comment:
As you advised, we showed and explained the phylum, class, order, and family level. The genus and species level data were moved to Supplementary Figure S4. We moved the description of the abundance values described in Results 2.4. to 2.2. Please refer to line 140-163. Thank you.
- Figure 3 is also listing all the alpha index. The finding should be the key to let reader to read and remember, not show all the data. Please only keep 1-2 panels, and remove them or as a supplementary figure.
Response to Comment:
As you advised, we moved some of the alpha indices to supplementary Figure S6, leaving only two alpha indices. Thank you.
- Figure 4. Please show the legend in group, and in sample.
Response to Comment:
As you advised, we added the legend in group, and in sample. Thank you.
- Figure 5. The comparing the tree? How to compare? Let the reads the read detail in figure itself?
Response to Comment:
As written in the figure legend, the color of the outer ring of the phylogenetic tree is the type of phylum, and the color of the inner circle is the type of the family. The distribution of microorganisms can be identified through the color composition and proportions of each location. You can also compare species diversity by comparing total species counts. The total number of species is described in results. Please refer to line 222-225. We improved the image quality so that the name of the species in the figure is visible. Thank you.
Round 2
Reviewer 1 Report
- Lines 17-18 “We found that FMT affected locomotion in three different ways: positive, none, and negative.” It is obvious that there are only three possible outcomes of FMT affecting locomotion. It is better to present the results for positive, none the same, and negative separately rather than writing the above sentence.
Response to Comment:
As you recommended, we corrected none to none the same. Please refer to line 18. Thank you.
Reply to Response:
Thank you for changing the sentence.
- There were 8 bacterial names in the keywords. It is better to delete these.
Response to Comment:
As you recommended, we removed the microbial name from the keywords. Please refer to line 29. Thank you.
Reply to Response:
Thank you for the modified manuscript.
- Could authors clarify the sex and number of the mice used in the experiment?
Response to Comment:
A total of 21 mice (15 females and 6 males) were used. There are 6 females and 1 male in the SL group, 4 females and 3 males in the ML group, 5 females and 2 males in the WL group. Please refer to line 90-91, line 101-103 and line 342. Thank you.
Reply to Response:
Thank you for your explanation. Could authors clarify the inconsistent number of male and female mice used in this study? The C57BL/6 mice used in this study were purchased from the Korea Basic Science Institute and would not be difficult to get. Moreover, I have a doubt about whether gender might affect the locomotion of the mice.
- In Figure 1 B, could authors add points and connecting lines, which would allow readers to more clearly compare the changes in the locomotion ability of mice before and after FMT.
Response to Comment:
As you suggested, we plotted the changes in locomotion ability of each mice before and after FMT for each group by adding dots and connecting lines. The graphs were added to the supplementary Figure S1. Thank you.
Reply to Response:
Thank you for modifying it. I would like to recommend replacing Figure 1 B with Figure S1.
- In Figure 1 B, ML group mice had lower hanging performances than the other two groups before FMT and remained lower after FMT. Does this indicate that the mice themselves, rather than the gut microbiome, determine their locomotion?
Response to Comment:
Even if multiple mice are subjected to FMT using the same feces, the same intestinal microbes will not colonize. In other words, we predicted that all mice would have different gut microbiome after FMTs. We also measured and grouped changes in locomotion recordings of the same mouse to exclude characteristics of the mouse itself.
In Fig. 1B, low locomotion records of the ML group before FMT could be attributed to the low locomotion of the mice themselves, whereas similar locomotion records after FMT were because of either not forming communities of locomotion regulating microbes or colonizing balanced gut microbes of both increasing and decreasing locomotion. Thank you.
Reply to Response:
Thank you for your explanation. Because mice exhibit coprophagy, mice in the same cage usually have a similar gut flora composition. Could the authors provide information on the cages?
- The mice were treated with cocktail of antibiotics for 7 days before FMT, and as seen in Figure 1 A the month 0-time point was having been treated with antibiotics. However, through Figures 2 and 3 the mice’s gut microbiome species and diversity were not affected by antibiotics before the FMT. Can the authors give any explanation?
Response to Comment:
There seems to be a misunderstanding in the interpretation of Figure 1A. 0-month is before antibiotic/antifungal treatment. After all experiments were performed at 0 months, the antibiotic mixture was ingested for 7 days, and FMT was performed for 3 months. Therefore, in Figure 2 and Figure 3, 0-month is not unaffected by antibiotics, but a normal condition before treatment with antibiotics. Although data were not shown, it was confirmed that the species and diversity of intestinal microorganisms significantly decreased after antibiotic/antifungal treatment.
To avoid misinterpretation of Figure 1A, the antibiotics treatment indication has been moved to the right. Thank you.
Reply to Response:
Thank you for the explanation and modified manuscript. I checked the changes and understood.
- Lines 363-365 “After completing FMT with the new human microbiome, the mice were kept in a controlled laboratory environment for an additional three months in the above normal mouse facility for 3 months” It appears that the mice were kept for an additional 3 months after 3 months of FMT, so it was a 6 months experiment. However, it looks like the mice were sacrificed at 3 months.
Response to Comment:
It means "during the 3 months the FMT is in progress", not "for 3 months after completing the FMT". The paragraph was deleted because it was incorrectly expressed and redundant. Thank you.
Reply to Response:
Thank you for the explanation and modified manuscript. I checked the changes and understood.
- Lines 128-134, sequencing methods were already written in the method section.
Response to Comment:
As you suggested, the redundant ‘sequencing methods’ were deleted. Thank you.
Reply to Response:
Thank you for modifying it.
- Mice were at least 15 months old after 3 months of FMT, and although they were not yet aged mice, 15-month-old mice already exhibited aging changes. In my opinion, it is necessary to set the non-FMT control group.
Response to Comment:
First of all, the subject of this thesis is not to look at changes due to aging, but to find human intestinal microbes that regulate movement. Second, individual characteristics due to aging would have been excluded through comparison before and after FMT of the same individual. Third, even if senescence was not excluded, senescence control would not have been considered because all mice experienced senescence equally. Thank you.
Reply to Response:
Thank you for your explanation.
- Lines 296-297, I don’t feel that this paper invented a new method for the gut microbiome.
Response to Comment:
There are many studies using FMT, but the method of research on intestinal microbes, which compares the same organism to exclude individual characteristics, is the first method developed by our research team. Thank you.’
Reply to Response:
Thank you for your explanation. I have a different opinion about the new method stated by the authors. To the best of my knowledge, there are many studies on human intestinal microbiota in mice transplanted. It is routine to treat mice with antibiotics prior to transplantation, and some studies used germ-free mice to transplant the human intestinal microbiome. I still do not feel that this paper invents a new method for the study of the gut microbiome, as FMT, antibiotic pre-treatment, and behavioral testing methods are already reported in existing studies. I strongly suggest that the authors could delete the sentence in the paper about the new method, such as lines 82, 85, and 292-293.
- Line 355, 10 healthy volunteers’ fecal samples were mixed in the optimized medium. Could authors give more information about the optimized medium?
Response to Comment:
The optimized medium is a culture medium supporting universal growth of intestinal microbes. We recently submitted a paper about the optimized medium to Nature. This paper is in review state. According to your comment, we added citation in line 351 and reference [53]. Thank you.
Reply to Response:
Thank you for your explanation. I found the reference [53] in research square (https://doi.org/10.21203/rs.3.rs-2649903/v1) and noticed that the method described in the optimized medium was the same as this article, “The fecal samples from 10 healthy volunteers were collected and mixed in the optimized medium to make mixed feces containing various human gut microbes. The fecal medium was created on the scheduled day for oral gavage and stored at room temperature in anaerobic containers. The mixed human feces medium was used to perform a FMT in mice. ”. Could authors give more information about the ingredients of the optimized medium? Authors may cite the preprint version in reference 53.
- Line 357, how long was the fecal medium stored at room temperature? How was the fecal medium preserved during the 3-month FMT?
Response to Comment:
As specified in the manuscript, fecal medium was freshly prepared every scheduled day for oral gavage and stored for 2 hours at room temperature before use. Because it was made fresh every FMT day during the 3-month, the fecal medium was discarded without preservation after use. Thank you.
Reply to Response:
Thank you for your explanation. The medium placed in an anaerobic chamber for 2 hours is not sufficient to reduce the oxygen in it. The medium needs to be placed in the anaerobic chamber for at least 12 hours to make the Resazurin indicator colorless. Moreover, could the authors provide information on the fecal collection and processing methods? Was the feces collected and centrifuged with PBS or saline?
Reference:
https://doi.org/10.1016/j.cell.2023.05.037s
- There are some spelling mistakes, for example, line 211 Clostridia requires italics, line 360 ul should be μL, line 389 65 ℃ should be 65℃, line 425 SILVA needs version number, line 442 “Analysis for Relative Abundance” should be “analysis for relative abundance”.
Response to Comment:
As you noted, we have changed italics for all taxonomic ranks in line 129-132, line 140-163 and line 266-268. We also changed ‘ul’ to ‘μL’ in line 355, ‘65 ℃’ to ‘65℃’ in line 380, and ‘Analysis for Relative Abundance’ to ‘analysis for relative abundance’ in line 433. We noted the SILVA version number (v138) in line 416. Thank you.
Reply to Response:
Thank you for modifying them. The name of the phylum does not need to be italicized. Moreover, the legend of Figure 1, line 116 “as the mean ± standard error of the mean (SEM)”, SEM should be SD.
Some editing of the English language is required for the manuscript.
Author Response
3. Could authors clarify the sex and number of the mice used in the experiment?
Response to Comment:
A total of 21 mice (15 females and 6 males) were used. There are 6 females and 1 male in the SL group, 4 females and 3 males in the ML group, 5 females and 2 males in the WL group. Please refer to line 90-91, line 101-103 and line 342. Thank you.
Reply to Response:
Thank you for your explanation. Could authors clarify the inconsistent number of male and female mice used in this study? The C57BL/6 mice used in this study were purchased from the Korea Basic Science Institute and would not be difficult to get. Moreover, I have a doubt about whether gender might affect the locomotion of the mice.
Response to Comment:
Initially, the experiment was started with the same number of male and female mice, but due to the characteristics of the behavior analysis experiment, mice that were not trained for exercise had to be excluded. Gender is not expected to have a significant effect on locomotion, given that no particular gender is biased toward one group. Thank you.
4. In Figure 1 B, could authors add points and connecting lines, which would allow readers to more clearly compare the changes in the locomotion ability of mice before and after FMT.
Response to Comment:
As you suggested, we plotted the changes in locomotion ability of each mice before and after FMT for each group by adding dots and connecting lines. The graphs were added to the supplementary Figure S1. Thank you.
Reply to Response:
Thank you for modifying it. I would like to recommend replacing Figure 1 B with Figure S1.
Response to Comment:
As you suggested, we changed the graph in Fig1B with the graph in FigS1. Thank you.
5. In Figure 1 B, ML group mice had lower hanging performances than the other two groups before FMT and remained lower after FMT. Does this indicate that the mice themselves, rather than the gut microbiome, determine their locomotion?
Response to Comment:
Even if multiple mice are subjected to FMT using the same feces, the same intestinal microbes will not colonize. In other words, we predicted that all mice would have different gut microbiome after FMTs. We also measured and grouped changes in locomotion recordings of the same mouse to exclude characteristics of the mouse itself.
In Fig. 1B, low locomotion records of the ML group before FMT could be attributed to the low locomotion of the mice themselves, whereas similar locomotion records after FMT were because of either not forming communities of locomotion regulating microbes or colonizing balanced gut microbes of both increasing and decreasing locomotion. Thank you.
Reply to Response:
Thank you for your explanation. Because mice exhibit coprophagy, mice in the same cage usually have a similar gut flora composition. Could the authors provide information on the cages?
Response to Comment:
Mice were individually managed, one per cage. Thank you.
10. Lines 296-297, I don’t feel that this paper invented a new method for the gut microbiome.
Response to Comment:
There are many studies using FMT, but the method of research on intestinal microbes, which compares the same organism to exclude individual characteristics, is the first method developed by our research team. Thank you.
Reply to Response:
Thank you for your explanation. I have a different opinion about the new method stated by the authors. To the best of my knowledge, there are many studies on human intestinal microbiota in mice transplanted. It is routine to treat mice with antibiotics prior to transplantation, and some studies used germ-free mice to transplant the human intestinal microbiome. I still do not feel that this paper invents a new method for the study of the gut microbiome, as FMT, antibiotic pre-treatment, and behavioral testing methods are already reported in existing studies. I strongly suggest that the authors could delete the sentence in the paper about the new method, such as lines 82, 85, and 292-293.
Response to Comment:
As you suggested, we deleted the sentence in the paper about the new method. Please refer to line 82-84 and line 289-291. Thank you.
11. Line 355, 10 healthy volunteers’ fecal samples were mixed in the optimized medium. Could authors give more information about the optimized medium?
Response to Comment:
The optimized medium is a culture medium supporting universal growth of intestinal microbes. We recently submitted a paper about the optimized medium to Nature. This paper is in review state. According to your comment, we added citation in line 351 and reference [53]. Thank you.
Reply to Response:
Thank you for your explanation. I found the reference [53] in research square (https://doi.org/10.21203/rs.3.rs-2649903/v1) and noticed that the method described in the optimized medium was the same as this article, “The fecal samples from 10 healthy volunteers were collected and mixed in the optimized medium to make mixed feces containing various human gut microbes. The fecal medium was created on the scheduled day for oral gavage and stored at room temperature in anaerobic containers. The mixed human feces medium was used to perform a FMT in mice. ”. Could authors give more information about the ingredients of the optimized medium? Authors may cite the preprint version in reference 53.
Response to Comment:
We explained the ingredients of the optimized medium in method part. Please refer to line 347-369. And we changed the reference [53] to the preprint journal you mentioned. Thank you.
12. Line 357, how long was the fecal medium stored at room temperature? How was the fecal medium preserved during the 3-month FMT?
Response to Comment:
As specified in the manuscript, fecal medium was freshly prepared every scheduled day for oral gavage and stored for 2 hours at room temperature before use. Because it was made fresh every FMT day during the 3-month, the fecal medium was discarded without preservation after use. Thank you.
Reply to Response:
Thank you for your explanation. The medium placed in an anaerobic chamber for 2 hours is not sufficient to reduce the oxygen in it. The medium needs to be placed in the anaerobic chamber for at least 12 hours to make the Resazurin indicator colorless. Moreover, could the authors provide information on the fecal collection and processing methods? Was the feces collected and centrifuged with PBS or saline?
Reference:
https://doi.org/10.1016/j.cell.2023.05.037s
Response to Comment:
Thank you for your comments. The collected feces were stored in an anaerobic chamber for 2 hours not for the purpose of completely reducing oxygen, but for the purpose of minimizing the loss of anaerobic microorganisms and storing them until all feces of 10 volunteers were collected.
We made every effort to minimize the loss of fecal microorganisms for FMT. The optimized medium was stored in an anaerobic chamber for more than 24 hours to remove oxygen before receiving feces, and FMT was performed as soon as possible after collecting feces. PBS or centrifugation treatment was not performed due to concerns about the loss of intestinal microbiome. However, in order not to cause abnormalities in the mouse's health, we finally chose to feed a small amount (20μL) intermittently through various trials. Thank you.
13. There are some spelling mistakes, for example, line 211 Clostridia requires italics, line 360 ul should be μL, line 389 65 ℃ should be 65℃, line 425 SILVA needs version number, line 442 “Analysis for Relative Abundance” should be “analysis for relative abundance”.
Response to Comment:
As you noted, we have changed italics for all taxonomic ranks in line 129-132, line 140-163 and line 266-268. We also changed ‘ul’ to ‘μL’ in line 355, ‘65 ℃’ to ‘65℃’ in line 380, and ‘Analysis for Relative Abundance’ to ‘analysis for relative abundance’ in line 433. We noted the SILVA version number (v138) in line 416. Thank you.
Reply to Response:
Thank you for modifying them. The name of the phylum does not need to be italicized. Moreover, the legend of Figure 1, line 116 “as the mean ± standard error of the mean (SEM)”, SEM should be SD.
Response to Comment:
As you noted, the name of the phylum written in italic has been replaced in line 127-130, line 139-141, line 150-152 and line 264-266. And we changed the ‘SEM’ to ‘SD’ in line 114. Thank you.

Reviewer 2 Report
The manuscript presented by the authors provides a comprehensive study on the “Identification of the intestinal microbes associated with loco-motion”, using a large and valuable dataset. The study is important in deepening our understanding of “mothur with gut microbiota”. After the modification, the paper has been improved. However, there are some issues not fully addressed in the manuscript.
Major reviews:
1. Your comments say you have using ImageGP to replot the figures. However, I'm not found you properly cited the tool. Please check it.
2. The figure 2 legend, you have labelled the level phylum, class, should remove the prefix such as “D_1_”, “D_2_”, “D_3_”, “D_4_” to make the legend clearer.
3. Figure 3, replace the “_” to blank it better to read. The x-axis label should be in 45 angle is easier to read.
4. Figure 4 have the same problem similar with figure, please replace the “_” to blank it better to read. The x-axis label should be in 45 angle is easier to read.
Author Response
1. Your comments say you have using ImageGP to replot the figures. However, I’m not found you properly cited the tool. Please check it.
Response to Comment:
According to your comment, we added citation in line 514 and reference [64]. Thank you.
2. The figure 2 legend, you have labelled the level phylum, class, should remove the prefix such as “D_1_”, “D_2_”, “D_3_”, “D_4_” to make the legend clearer.
Response to Comment:
As you advised, we have removed the unnecessary prefix in figure 2. Thank you.
3. Figure 3, replace the “_” to blank it better to read. The x-axis label should be in 45 angle is easier to read.
Response to Comment:
As you advised, we replaced the "-" to blank, and made the x-axis labels 45 angles in figure 3. Thank you.
4. Figure 4 have the same problem similar with figure, please replace the “_” to blank it better to read. The x-axis label should be in 45 angle is easier to read.
Response to Comment:
As you advised, we replaced the "-" to blank, and made the x-axis labels 45 angles in figure 4. Thank you.

Round 3
Reviewer 1 Report
After carefully reviewing the revised manuscript and the authors' response. I found that all of the concerned questions were well answered and this paper met the criteria for acceptance. I recommend this manuscript could be accepted and published.
Author Response
Thank you for your positive review.
Reviewer 2 Report
The author's response has answered my suggestion. I agree with the publication of this article.
Author Response
Thank you for your positive review.